# Mental Health of the Russian Federation Population versus Regional Living Conditions and Individual Income

**DOI:** 10.3390/ijerph20115973

**Published:** 2023-05-27

**Authors:** Sergey A. Maksimov, Marina B. Kotova, Liliya I. Gomanova, Svetlana A. Shalnova, Yulia A. Balanova, Svetlana E. Evstifeeva, Oksana M. Drapkina

**Affiliations:** National Medical Research Center for Therapy and Preventive Medicine, Ministry of Healthcare of the Russian Federation, Petroverigsky Lane 10 bld., 101990 Moscow, Russia; m1979sa@yandex.ru (S.A.M.); mari-115@rambler.ru (M.B.K.); svetlanashalnova@yandex.ru (S.A.S.); jbalanova@gnicpm.ru (Y.A.B.); sevstifeeva@gnicpm.ru (S.E.E.); drapkina@bk.ru (O.M.D.)

**Keywords:** mental health, stress, anxiety, depression, living environment, regional characteristics, Russia

## Abstract

The objective of our study was to assess the impact of regional living conditions on the Russian population’s mental health. For the analysis, we used data from the cross-sectional stage of a 2013–2014 study, “Epidemiology of Cardiovascular Diseases in the Regions of the Russian Federation (ESSE-RF)”. The final sample included 18,021 men and women 25–64 years of age from 11 regions of Russia. Using principal component analysis, we performed an integral simultaneous assessment of stress, anxiety, and depression. To describe the regional living conditions, we utilized five regional indices, which were computed from publicly available data of the Federal State Statistics Service of Russia. Overall, mental health indicators were improved, on the one hand, with the deterioration of social conditions and an aggravation of the demographic depression in the region, but on the other hand, they were improved with an increase in economic and industrial development, along with economic inequality among the population. In addition, the impact of regional living conditions on mental health increased with a higher individual wealth. The obtained results provided new fundamental knowledge on the impact of the living environment on health, using the case study of the Russian population, which has been little studied in this regard.

## 1. Introduction

Mental health is an integral part of overall health and well-being. In addition, it is a fundamental human right. According to the World Health Organization’s (WHO) estimates for 2019, 15% of the working-age adult population suffered from mental disorders, of which 280 million were depressed (including 23 million children and youths) [1]. Every 13th person worldwide lives with the most common anxiety disorders [2], and their prevalence in the population of different countries varies from 0.6% to 10.4% [3].

Large studies and meta-analyses discovered that mental health indicators, such as stress, anxiety, and depressive disorders, were associated with major adverse cardiovascular events, hospital readmission, and death, independent of traditional risk factors [2,4,5,6]. The Global Burden of Diseases, Injuries, and Risk Factors Study of 2017 showed that mental disorders were among the top three causes of disability [3]. Furthermore, individuals experiencing chronic stress and increased levels of anxiety and depression were more susceptible to behavioral risk factors for noncommunicable diseases, primarily smoking, alcohol consumption, poor diet, and lack of physical exercise [7]. The side effects associated with medicamentous therapy in the treatment of psychiatric disorders also play a role in premature mortality and contribute to the development of obesity, hyperglycemia, and dyslipidemia [4].

Mental health largely depends on the individual characteristics of a person: primarily on gender, age, marital status, and socioeconomic status. For instance, anxiety and depression are almost twice as common among women [8,9], and this gender gap is believed to be associated with sex differences in biological and psychological susceptibility, as well as environmental factors acting at both the microlevel and macrolevel [10,11]. The peak prevalence of mental health disorders in both sexes occurs in the second and third decades of their lives [10,11]. As for marital status, single people, especially women, as well as those living separately or divorced, experience significantly higher rates of depression [9,12]. Many studies demonstrated an inverse relationship between socioeconomic status and mental health [13,14]. At the same time, the protective factors for mental health are not so much the absolute income size as the availability of savings, real estate, and financial stability [15,16]. It is the presence of perceived wealth that is associated with a decrease in symptoms of depression, anxiety, and stress [12,15,17].

At the same time, health in general depends not only on individual characteristics but also on the living environment. The same is true for mental health. For example, the WHO World Mental Health Survey Initiative reported that the prevalence of major depressive disorder was higher in high-income countries (France, the Netherlands, New Zealand, and the USA), while the lowest rates were detected in India (Puducherry), Mexico, China (Shenzhen), and South Africa [9]. A later study by F.C. Shahbazi et al., conducted in 181 countries, also pointed to an inverse relationship between socioeconomic inequality and the prevalence of depressive symptoms [18].

However, it is clear that the geographic location is not, by and large, a predictor of differences in mental health. The systemic socioenvironmental approach that takes into account the interaction of both individual and environmental socioeconomic factors of health proposes that the so-called social production of diseases plays a significant role [19,20]. The theoretical concept of the impact of the living environment is based on the fact that the fundamental characteristics of life (e.g., socioeconomic inequality) affect intermediate factors, which, in turn, affect psychoemotional and behavioral predictors. The latter are direct or indirect health risk factors [20].

There are many studies evaluating the impact of the regional living environment at the neighborhood level (i.e., at the small spatial scale) [21]. Their results primarily indicate the favorable impact of the high socioeconomic status of the residential neighborhood on mental health. At the same time, there are few studies evaluating the impact of the living environment on the likelihood of mental disorders at the level of much larger areas. These studies yielded conflicting results: some of them implied a direct impact of the socioeconomic characteristics in the residential neighborhoods on mental health [22], while other studies proposed the presence of an inverse relationship [23,24,25] or suggested no impact whatsoever [26]. The inconsistency of these results was probably associated, among other things, with the cultural differences between different populations and the need for additional research in different countries and on populations with different mindsets. According to epidemiological studies, the prevalence of increased anxiety and depression in Russia was 46.3% and 25.6%, respectively, while the clinical level of depression and anxiety disorders reached 8.8% and 18.1%, correspondingly [27]. In this study, the impact of individual characteristics on mental health was analyzed; however, the effect of living conditions was not examined. In this regard, the objective of our study was to assess the impact of regional living conditions on the mental health (stress, anxiety, and depression) in the Russian Federation population. Furthermore, our additional task was to analyze the interaction of the effects of regional and individual characteristics. The research hypothesis is the assumption that the living conditions of the population at the level of large regions have both an independent influence and influence in interaction with individual social characteristics on the individual mental health of Russians.

## 2. Materials and Methods

### 2.1. Study Design and Sampling

We used cross-sectional data from the study “Epidemiology of Cardiovascular Diseases in the Regions of the Russian Federation (ESSE-RF)”, conducted in 2013–2014 [28]. In total, 21,923 individuals aged 25–64 years old from 13 regions of the Russia participated in the study. To form the sample, the Kish grid was used, which provides a systematic multistage random sample according to the district principle in medical organizations. Ethical committees of three scientific and medical centers approved the study: National Medical Research Center for Therapy and Preventive Medicine, Russian Cardiology Research and Production Complex and V.A. Almazov Federal Medical Research Center. The principles of Good Clinical Practice (GCP) and the Declaration of Helsinki were applied during the study. Prior to the study, all participants were informed of their rights and the features of the survey and questionnaires, and they signed informed consent. In total, about 80% of those invited agreed to take part in the study, with little variation by region.

Individuals with missing data on depression, stress, anxiety, and individual characteristics (*n* = 898 individuals), as well as on smoking, were excluded from the final sample. A subsample of the city of St. Petersburg (*n* = 1460 people) was excluded from the study. The fact is that St. Petersburg is only a small, highly urbanized territory, while the remaining 12 regions are large territories, with characteristics of both urban and rural settlements. Furthermore, a subsample of the Orenburg Oblast (*n* = 1544 people) was excluded from the analysis due to the insufficient quality of data on depression, stress, and anxiety. The final sample was 18,021 individuals from 11 regions (6831 men and 11,190 women).

### 2.2. Methods for Assessing Depression, Stress and Anxiety

We assessed the level of anxiety and depression using the Hospital Anxiety and Depression Scale (HADS) that was validated for Russia [29]. The Perceived Stress Scale [30] was employed to determine susceptibility to stress. The scores obtained by either scale are quantitative indicators: for depression and anxiety, the score ranges from 0 to 21 points, whereas for stress, the score is in the range of 0–40 points. An increase in score is interpreted as an increase in depression, anxiety, and stress.

Since the stress, depression, and anxiety scores are, to a large extent, interrelated (correlation coefficients of 0.48–0.57), we carried out an integral simultaneous assessment of these three indicators, altogether considered by us as a mental health scale (MHS). To fulfill such an assessment, we used the principal component method for three indicators to identify a latent variable with a high load on depression (0.81), stress (0.82), and anxiety (0.86) and providing an explanation for 69.4% of the variance. For each participant in our study, we calculated an individual predisposition to a latent factor with a normal distribution, a mean of 0, and a standard deviation of 1. An increase in the MHS value implied the deterioration of the mental health based on the combination of depression, stress, and anxiety.

### 2.3. Individual Characteristics

Through face-to-face interviews, a number of individual characteristics of study participants were determined: gender, age, employment, level of education (higher/other than higher), marital status (married/single), place of residence (urban/rural), and attitudes towards smoking (never, quit, or currently smokes).

The level of income was assessed indirectly by the answer to the question, “How do you evaluate the well-being of your family compared to others?” The answer options “very poor” and “relatively poor” were rated as low income. The answer option “average (not rich, albeit not poor)” was interpreted as middle income. The answer options “relatively wealthy” and “very wealthy” were regarded as high income.

### 2.4. Regional Characteristics

When assessing the characteristics of the regions that took part in the study, we used previously developed multicomponent regional indices. A detailed description of the indices is presented in another publication [31]. Briefly, regional characteristics from the official website of the Federal State Statistics Service of Russia (www.gks.ru, accessed on 10 April 2023) for 2010–2014 were used. Then, using the principal component analysis, we identified five multicomponent indices and characterized different features of population life in the regions: socio-geographical, demographic, industrial, economic, and mixed. The increase in the socio-geographical index reflects the deterioration of climatic living conditions (the more northerly location of the region), as well as the deterioration of the social environment, including an increase in alcohol consumption per capita, crime, worsening living conditions, and conditions for school students. An increase in the demographic index reflects an increase in the depopulation of the region’s population with a characteristic decrease in the total fertility rate and natural population growth against changes in the population age structure in favor of older age groups. The industrial index characterizes the development of the mining and energy industries in the region, an increase in the level of exposure of population to adverse anthropogenic effects, and increase in mortality (infectious diseases, tuberculosis, injuries, and accidents). The increase in the economic index characterizes the growth of trade, income, and consumption per capita, as well as the increase in economic inequality in the region. The mixed index is the most difficult to interpret, since, in general, it does not allow for an unambiguous characterization based on the entirety of its constituent indicators. However, for data completeness, we describe the associations obtained for this index.

### 2.5. Statistical Data Processing

When describing variables, quantitative indicators are presented as the mean and its standard deviation, while qualitative indicators are presented as frequencies. Pearson’s chi-squared test was used to evaluate qualitative differences. Since the distribution of quantitative indicators does not correspond to a normal one, the nonparametric Kruskal–Wallis test was used to assess their differences. Since the analytical data set is a two-level sample (study participants in the regions), generalized estimating equations (GEE) with robust standard errors were used to measure associations. Depression, stress, anxiety, and MHS are quantitative indicators, so their relationship with regional indices was assessed using linear regression (GEE). All regression models were adjusted for individual characteristics: gender, age, employment, urban/rural, education, marital status, smoking, and income level. All regional indices were entered into the regression models simultaneously. The statistical analysis revealed some interactions of gender with regional indicators; therefore, in addition to analyses of the overall sample, we also performed separate analyses for low-, middle-, and high-income individuals. To compare the importance of regional conditions with individual predictors of mental health, effect parameter values were calculated using Wald’s chi-square from regression models. A value of 0.05 was taken as the critical level of statistical significance. The statistical analysis was carried out in the program SPSS 22 (IBM Corporation, Armonk, NY, USA).

## 3. Results

The main characteristics of the sample are presented in Table 1. In the structure of the general sample, urban residents, women, married individuals, study subjects without higher education, professionally employed people, and nonsmokers were represented by larger shares. The mean age was 46.5 ± 11.6 years. The mean scores for depression, stress, and anxiety were 5.0 ± 3.7, 15.1 ± 6.3, and 6.2 ± 3.9 points, respectively. The income was evaluated as low, middle, or high by 2043 (11.3%), 14,043 (77.9%), and 1935 (10.8%) study participants, correspondingly. Virtually all individual and regional indicators (with the exception of the industrial and mixed indices) differed in a statistically significant way between income-based groups. For most indicators, the following linear trends with an increase in income were characteristic: a reduction in the shares of women and nonsmokers, and a decrease in mean age and scores for all mental health indicators; on the contrary, there was an increase in the shares of married people, study subjects with higher education, and professionally employed individuals, as well as an increase in the socio-geographical index.

Associations of depression with individual and regional factors are presented in Table 2. In the general sample, the depression score declines with the deterioration of the social environment (socio-geographical index: B = −0.781, *p* < 0.001), as well as with the growth of regional economic development, income, and economic inequality of the population (economic index: B = −0.423, *p* = 0.005). In addition, we observed a direct association of depression with the mixed index (B = 0.139, *p* = 0.005). With an increase in individual income, the number of associations of depression with regional predictors increased. Depression was associated solely with the socio-geographical index in the low-income group, while associations with the economic and mixed indices were observed as well in the middle-income and high-income groups. Moreover, in the high-income group, there was an inverse association between the demographic index and depression (B = −0.402, *p* < 0.001)—i.e., with a growing demographic depression in the region, the depression score declined.

Associations of stress with individual and regional factors are presented in Table 3. In the general sample, the stress score declined with the deterioration of the social environment (B = −1.276, *p* < 0.001) but increased with the growth of the mixed index (B = −0.292, *p* = 0.001). Similar to depression, the number of associations of stress with regional indices increased with individual income. Stress was associated solely with the socio-geographical index in the low-income group, while we observed an additional association with the mixed index in the middle-income and high-income groups. In the high-income group, we revealed an inverse association with the industrial development of the region (industrial index, B = −0.421, *p* = 0.008).

Associations of anxiety with individual and regional factors are presented in Table 4. In the general sample, the value of anxiety decreased with the deterioration of the social environment (B = −0.366, *p* = 0.043), but it increased with the growth of the mixed index (B = 0.206, *p* < 0.001). Individuals with low incomes did not have associations of anxiety with regional indices. For people with a middle income, the value of anxiety increased with the growth of the regional economic development, income size, and economic inequality of the population (economic index: B = −0.428, *p* = 0.040), and it was also directly associated with the mixed index. The largest number of associations was observed in people with high income: as in the general sample, anxiety was statistically significantly associated with the socio-geographical and mixed indices, but there were additional statistically significant inverse associations with the demographic (B = −0.213, *p* = 0.014) and industrial (B = −0.166; *p* = 0.018) indices.

Associations of MHS scores with individual and regional factors are presented in Table 5. The improvement in MHS scores was associated with a growth in the socio-geographical (B = −0.203; *p* < 0.001) and economic (B = −0.108; *p* = 0.045) indices, as well as with a decrease in the mixed index (B = 0.055; *p* < 0.001). As individual income increased, so did the number of associations. If in the group of people with low income there was an association solely with the socio-geographical index, and then in the middle-income group, we observed already all three associations similar to the general sample (with the socio-geographical, mixed, and economic indices). In the group of high-income individuals, in addition to associations with the socio-geographical and mixed indices, we established inverse associations with the demographic (B = −0.067, *p* = 0.001) and industrial (B = −0.045, *p* = 0.003) indices.

In the general sample, the significance of the contribution of regional indices to depression, stress, anxiety, and MHS did not reach the level of individual predictors, such as gender, educational attainment level, income size, and professional employment (Table 6). The significance of the contribution for these individual characteristics reached 100 or even 200 for some indicators. However, the significance of the contribution of the socio-geographical and mixed indices was consistent and quite high, varying from 4 to 40. In addition, the economic index exhibited a contribution of 8.1 to the indicators of depression. Demographic and industrial indices in the general sample were characterized by low contributions to mental health indicators. In general, the analysis of the contribution of predictors indicates a greater significance of individual socio-demographic characteristics in mental health. Regional living conditions make a smaller contribution; however, in some cases, it is statistically significant.

Hence, the results of our study confirmed a significant contribution of regional living conditions to mental health indicators, both in general and in particular (i.e., in terms of individual indicators of depression, stress, and anxiety). The most pronounced and consistent impact was demonstrated by the social conditions (socio-geographical index), as well as the mixed index. Unfortunately, the latter was difficult to interpret. Moreover, for the depression and mental health scores in general, the residential economic conditions were of significant importance. Industrial development and the level of regional demographic depression in the general sample did not exhibit any consistent effect; however, statistically significant associations were observed in some income subgroups. The direction of associations of regional living conditions with mental health turned out to be the reverse (with the exception of the mixed index), which was rather unexpected. That is to say, the mental health improved, on the one hand, with the deterioration of social conditions and an increase in the demographic depression of the region, but, on the other hand, with an increase in economic and industrial development, as well as economic inequality of the population.

## 4. Discussion

In our opinion, the results of studies of the impact of the environment on mental health substantially depend on national cultural idiosyncrasies, and this explains such contradictory results of our research and other similar studies. For example, a Korean study conducted in two districts of Seoul showed that the prevalence of depression in a less deprived area was higher than in an area with a low socioeconomic status (23.1% and 16.3%, respectively) [22]. At the same time, a study conducted in 25 provinces in China demonstrated the opposite results: people living in provinces with higher socioeconomic status were more likely to report a low depression level, regardless of individual characteristics [23]. Similar results, but with some stress, were demonstrated by H. Wang based on an analysis of a sample of residents of 21 Chinese cities [24]. In an American study, the effect of territorial features was assessed simultaneously at two levels. At the county level, social capital independently predicted the likelihood of depression, while at the state level, it mediated the relationship of depression with income inequality [25].

The results of our study imply that the mechanisms of influence of environmental factors on mental health are quite complex and largely interrelated; however, the foremost impact of the social characteristics of living conditions (socio-geographical index) is obvious. Presumably, with the deterioration of the social environment, the subjective level of individual requirements for education, career, financial well-being, and marital relationship stability decreases. Sensu A. Maslow’s hierarchy of needs for people living in regions with a low-quality social environment makes no sense to satisfy such requirements as aesthetic and cognitive needs, along with the needs for self-actualization and respect, until the basic needs of the lower hierarchical levels are satisfied. Accordingly, the state of mental health is not subject to strain, which does not lead to an increase in symptoms of depression and anxiety, as well as to the progression of stress. Also important is social support and adaptation, which, for example, are lower in socially developed urbanized cities. This indirectly affects the state of mental health [32]. Living in large cities, on the one hand, is associated with more developed socioeconomic infrastructure, which increases the likelihood of interaction with a large number of people; but on the other hand, this leads to potential mental overload and emotional overstrain [33,34]. Contrariwise, other data suggest a protective role of social urbanization in relation to mental health [35] or a lack of such an association [36].

The stratification of the sample, based on individual characteristics, revealed another noteworthy feature: with an increase in individual wealth, the impact of regional living conditions on mental health increased. While individuals with low individual incomes were characterized by associations solely with the socio-geographical index, people of middle-level income also had associations with the mixed and economic indices. The maximum number of associations, including the industrial and demographic indices, was observed among people with high individual wealth. This relationship can be explained by the strain and anxiety of people with a high level of prosperity regarding their social and financial position in society, which primarily depends on the socioeconomic development in the region. In regions with a low socioeconomic level, it is easier for people with a high material prosperity to achieve their goals due to low competition and reduced prices for goods and services. Chinese scientists came to the opposite conclusion, finding that, with an increase in individual socioeconomic status, the effect of regional deprivation on depressive symptoms declined. The authors explained this by the fact that people with higher socioeconomic status had greater access to healthcare services, and therefore, they were less dependent on services provided at the community level [37]. At the same time, a number of studies demonstrated the lack of interaction between the regional and individual socioeconomic levels [38,39].

For the first time in medical science, this study presents data on the association of mental health with the conditions of the living environment in the population of the Russian Federation. In addition, this is one of the few studies worldwide on individual mental health that assessed the impact of living conditions at the level of large territorial units within the same country. As mentioned above, the vast majority of similar studies operated on a smaller spatial scale. At the same time, the spatial scale, which characterizes the impact of a living environment, is important because different factors may affect health indicators at different spatial levels, up to inversion. A striking example is the well-known phenomenon in epidemiology called the “Swiss paradox”: at a different spatial scale, income inequality can have both direct and inverse effects on population health [40]. While most international studies confirmed a negative impact of growing income inequality on population health, when measured at a more specific (smaller) geographic level, it has no effect or is even associated with better health outcomes. Indicative in this regard is the American study on the dependence of obesity on income inequality at two spatial levels: a direct relationship was noted at the county level, while at the state level, there was no association whatsoever [41].

In general, the results characterize the theoretical patterns of individual health formation depending on the living conditions of the population. At the same time, study results allow us to look at aspects of the population mental health formation from a new perspective. Quite complex patterns of mental health formation, which, on the one hand, improves with the deterioration of social conditions and an increase in the demographic depression of the region and, on the other hand, improves with an increase in economic and industrial development, as well as economic inequality of the population, indicate the need for differentiated approaches to the mental-health-disorder prevention depending on the environmental characteristics. Study results indicate a rather high psychological stability of the Russian population in an unfavorable social and demographic environment. At the same time, the regional development both in the economy and industry leads to a decrease in the psychological stability of the population. This requires the development of appropriate programs for the population prevention and adaptation.

Along with the advantages, there are some limitations to our study. First, it is necessary to admit that its cross-sectional design limits evidence in terms of causation. Moreover, the results of our study are fairly difficult to compare with those of similar studies, due to differences—which are sometimes quite significant—in the methodology used for determining both exposure (characteristics of the regions of residence) and outcome (indicators of stress, anxiety, and depression). There is no gold standard in the practice of epidemiological research, and the applicability of certain methodological approaches is determined by the available data and the objective of the study. This issue substantially limits the interpretation of the obtained results. As a limitation, it should be noted that the study analyzed epidemiological monitoring data for 2013–2014. It is highly likely that the global COVID-19 epidemic accompanied by both changes in the population living conditions and a significant impact on mental health could change the identified associations. In this regard, it would be reasonable to compare data with similar data in the COVID-19 period and post-COVID-19 period.

## 5. Conclusions

Hence, our study allowed us to assess the individual dependence of mental health on the characteristics of Russian regions, primarily on social living conditions and, to a lesser extent, on demographic conditions, along with an economic and industrial development. The contributions of regional characteristics to mental health indicators are lower than that of some well-known individual characteristics, such as gender, age, and individual socioeconomic status (income, education, and employment), albeit quite significant. In addition, we revealed the interaction between regional conditions and individual income in their cumulative impact on mental health. This interaction is manifested by a decrease in the resistance of mental health to changes in environmental conditions with an increase in individual income. Our results provide novel fundamental knowledge on the impact of the living environment on health, using the case study of the Russian population, which has been little studied in this regard. For the healthcare system, the obtained results provide useful information on the need to use differentiated approaches to the prevention of mental health disorders, depending on the characteristics of the living environment. For government agencies, the results of our study show the direction for the socioeconomic development of the region (city, oblast), focused on maintaining the health of the population, both physical and mental.

## Figures and Tables

**Table 1 ijerph-20-05973-t001:** The main characteristics of the study sample.

Characteristics		Entire Sample, *n* = 18,021	Income Level Groups			*p*-Value
			Low, *n* = 2043	Middle, *n* = 14,043	High, *n* = 1935	
Place, urban		77.6% (13,990)	79.8% (1631)	77.3% (10,849)	78.0% (1510)	0.030
Sex, women		62.1% (11,190)	71.0% (1450)	62.4% (8766)	50.3% (974)	<0.001
Marital status, married		64.9% (11,693)	52.1% (1064)	66.0% (9273)	70.1% (1356)	<0.001
Education, higher		42.2% (7607)	31.4% (641)	41.8% (5868)	56.7% (1098)	<0.001
Employment, employed		75.3% (13,561)	60.3% (1232)	76.4% (10,727)	82.8% (1602)	<0.001
Smoking	Non-smoker	61.4% (11,066)	63.0% (1287)	62.0% (8707)	55.4% (1072)	<0.001
	Quit	17.0% (3063)	16.0% (326)	16.6% (2327)	21.2% (410)	
	Smoker	21.6% (3892)	21.0% (430)	21.4% (3009)	23.4% (453)	
Age		46.5 ± 11.6	50.2 ± 10.7	46.4 ± 11.6	43.4 ± 11.5	<0.001
Socio-geographical index		0.054 ± 0.956	−0.151 ± 0.987	0.066 ± 0.949	0.186 ± 0.940	<0.001
Demographic index		0.041 ± 1.014	0.185 ± 1.012	0.020 ± 1.018	0.041 ± 0.971	<0.001
Industrial index		−0.087 ± 0.960	−0.132 ± 0.849	−0.091 ± 0.963	−0.014 ± 1.040	0.077
Mixed index		0.058 ± 1.095	−0.018 ± 0.988	0.067 ± 1.101	0.0731 ± 1.158	0.10
Economic index		0.059 ± 0.940	−0.001 ± 0.973	0.084 ± 0.943	−0.061 ± 0.869	<0.001
Depression		5.0 ± 3.7	6.8 ± 3.8	4.9 ± 3.6	4.1 ± 3.4	<0.001
Stress		15.1 ± 6.3	18.4 ± 6.7	14.8 ± 6.1	13.5 ± 6.1	<0.001
Anxiety		6.2 ± 3.9	7.8 ± 4.2	6.1 ± 3.8	5.5 ± 3.7	<0.001
Mental health		−0.031 ± 1.007	0.533 ± 1.056	−0.075 ± 0.978	−0.302 ± 0.945	<0.001

**Table 2 ijerph-20-05973-t002:** Associations of depression with individual and regional factors.

Predictors		Entire Sample, *n* = 18,021		Income Level Groups					
				Low, *n* = 2043		Middle, *n* = 14,043		High, *n* = 1935	
		B-Coefficient	*p*-Value	B-Coefficient	*p*-Value	B-Coefficient	*p*-Value	B-Coefficient	*p*-Value
Individual predictors									
Rural place of residence (ref.: urban)		−0.118	0.42	−0.133	0.48	−0.101	0.52	−0.252	0.28
Men (ref.: women)		−0.769	<0.001	−0.756	0.023	−0.788	<0.001	0.039	<0.001
Marital status (ref.: single)		−0.150	0.005	−0.353	0.027	−0.127	0.063	−0.078	0.61
Higher education (ref.: other than higher)		−0.620	<0.001	−0.711	0.001	−0.578	<0.001	−0.927	<0.001
Employed (ref.: unemployed)		−0.594							
Smoking (ref.: non-smoker)	Quit	−0.409	0.006	−0.324	0.19	−0.428	0.012	−0.369	0.12
	Smoker	−0.113	0.41	0.018	0.95	−0.096	0.52	−0.318	0.23
Income (ref.: high)	Middle	0.351	0.009	–	–	–	–	–	–
	Low	1.659	<0.001	–	–	–	–	–	–
Age		0.045	<0.001	0.037	0.005	0.048	<0.001	0.039	<0.001
Regional predictors									
Socio-geographical index		−0.781	<0.001	−0.769	<0.001	−0.789	<0.001	−0.737	<0.001
Demographic index		−0.123	0.19	0.074	0.62	−0.120	0.20	−0.402	<0.001
Industrial index		0.108	0.22	0.137	0.35	0.124	0.17	−0.007	0.91
Mixed index		0.139	0.005	−0.114	0.33	0.165	0.001	0.152	<0.001
Economic index		−0.423	0.005	−0.250	0.20	−0.481	0.002	−0.204	0.012

**Table 3 ijerph-20-05973-t003:** Associations of stress with individual and regional factors.

Predictors		Entire Sample, *n* = 18,021		Income Level Groups					
				Low, *n* = 2043		Middle, *n* = 14,043		High, *n* = 1935	
		B-Coefficient	*p*-Value	B-Coefficient	*p*-Value	B-Coefficient	*p*-Value	B-Coefficient	*p*-Value
Individual predictors									
Rural place of residence (ref.: urban)		0.117	0.54	−0.365	0.40	0.207	0.24	−0.275	0.50
Men (ref.: women)		−2.128	<0.001	−2.137	<0.001	−2.177	<0.001	−1.810	<0.001
Marital status (ref.: single)		−0.323	<0.001	−0.385	0.28	−0.313	<0.001	−0.288	0.33
Higher education (ref.: other than higher)		−0.548	<0.001	−0.909	0.014	−0.497	<0.001	−0.790	0.004
Employed (ref.: unemployed)		−0.616	<0.001	−0.790	0.013	−0.602	<0.001	−0.695	0.023
Smoking (ref.: non-smoker)	Quit	0.021	0.88	0.198	0.67	−0.063	0.71	0.374	0.14
	Smoker	0.272	0.14	0.397	0.34	0.231	0.21	0.440	0.19
Income (ref.: high)	Middle	0.914	<0.001	–	–	–	–	–	–
	Low	3.850	<0.001	–	–	–	–	–	–
Age		−0.003	0.66	−0.009	0.65	−0.004	0.62	0.006	0.71
Regional predictors									
Socio-geographical index		−1.276	<0.001	−1.019	0.005	−1.341	<0.001	−1.018	<0.001
Demographic index		0.188	0.46	0.117	0.69	0.218	0.40	−0.025	0.91
Industrial index		0.087	0.71	0.134	0.65	0.158	0.51	−0.421	0.008
Mixed index		0.292	0.001	0.141	0.55	0.299	0.001	0.344	<0.001
Economic index		−0.423	0.14	−0.288	0.43	−0.510	0.072	0.153	0.44

**Table 4 ijerph-20-05973-t004:** Associations of anxiety with individual and regional factors.

Predictors		Entire Sample, *n* = 18,021		Income Level Groups					
				Low, *n* = 2043		Middle, *n* = 14,043		High, *n* = 1935	
		B-Coefficient	*p*-Value	B-Coefficient	*p*-Value	B-Coefficient	*p*-Value	B-Coefficient	*p*-Value
Individual predictors									
Rural place of residence (ref.: urban)		−0.040	0.87	−0.304	0.36	−0.009	0.97	−0.049	0.82
Men (ref.: women)		−1.890	<0.001	−2.104	<0.001	−1.892	<0.001	−1.733	<0.001
Marital status (ref.: single)		−0.140	0.002	−0.079	0.69	−0.149	0.001	−0.094	0.24
Higher education (ref.: other than higher)		−0.354	<0.001	−0.448	0.048	−0.346	<0.001	−0.411	0.043
Employed (ref.: unemployed)		−0.674	<0.001	−1.059	<0.001	−0.627	<0.001	−0.602	0.025
Smoking (ref.: non-smoker)	Quit	0.078	0.56	0.339	0.10	0.004	0.97	0.260	0.18
	Smoker	−0.018	0.87	−0.378	0.23	−0.004	0.97	0.227	0.41
Income (ref.: high)	Middle	0.203	0.20	–	–	–	–	–	–
	Low	1.416	<0.001	–	–	–	–	–	–
Age		0.013	0.023	−0.006	0.60	0.014	0.009	0.017	0.084
Regional predictors									
Socio-geographical index		−0.366	0.043	−0.345	0.12	−0.363	0.056	−0.354	<0.001
Demographic index		−0.087	0.52	−0.118	0.49	−0.074	0.59	−0.213	0.014
Industrial index		0.064	0.64	0.105	0.52	0.086	0.54	−0.166	0.018
Mixed index		0.206	0.001	0.120	0.46	0.225	<0.001	0.145	<0.001
Economic index		−0.335	0.11	0.059	0.79	−0.428	0.040	−0.078	0.58

**Table 5 ijerph-20-05973-t005:** Associations of mental health with individual and regional factors.

Predictors		Entire Sample, *n* = 18,021		Income Level Groups					
				Low, *n* = 2043		Middle, *n* = 14,043		High, *n* = 1935	
		B-Coefficient	*p*-Value	B-Coefficient	*p*-Value	B-Coefficient	*p*-Value	B-Coefficient	*p*-Value
Individual predictors									
Rural place of residence (ref.: urban)		−0.009	0.83	−0.070	0.34	0.001	0.98	−0.049	0.44
Men (ref.: women)		−0.417	<0.001	−0.439	<0.001	−0.423	<0.001	−0.366	<0.001
Marital status (ref.: single)		−0.051	<0.001	−0.070	0.20	−0.049	<0.001	−0.036	0.25
Higher education (ref.: other than higher)		−0.139	<0.001	−0.181	0.005	−0.130	<0.001	−0.193	0.001
Employed (ref.: unemployed)		−0.174	<0.001	−0.247	<0.001	−0.168	<0.001	−0.144	0.001
Smoking (ref.: non-smoker)	Quit	−0.034	0.30	0.014	0.80	−0.049	0.18	0.012	0.77
	Smoker	0.003	0.93	−0.014	0.87	0.004	0.91	0.018	0.78
Income (ref.: high)	Middle	0.117	0.006	–	–	–	–	–	–
	Low	0.570	<0.001	–	–	–	–	–	–
Age		0.006	<0.001	0.003	0.80	0.006	<0.001	0.006	0.008
Regional predictors									
Socio-geographical index		−0.203	<0.001	−0.183	0.002	−0.207	<0.001	−0.181	<0.001
Demographic index		−0.011	0.77	0.003	0.96	−0.007	0.85	−0.067	0.001
Industrial index		0.024	0.50	0.034	0.47	0.032	0.38	−0.045	0.003
Mixed index		0.055	<0.001	0.009	0.83	0.060	<0.001	0.053	<0.001
Economic index		−0.108	0.045	−0.039	0.56	−0.129	0.017	−0.021	0.47

**Table 6 ijerph-20-05973-t006:** The meaning of the model effect criteria from regression models (Likelihood Type III test and chi-square Wald).

Characteristics	Depression	Stress	Anxiety	Mental Health
Individual predictors				
Place	0.6	0.4	0.1	0.1
Sex	27.4	70.0	150.6	83.5
Marital status	8.0	15.9	9.6	18.2
Education	210.6	18.2	19.6	59.6
Employment	51.4	24.5	38.1	48.9
Smoking	26.4	2.6	0.7	2.7
Income	114.9	185.1	67.5	154.2
Age	121.3	0.2	5.1	19.2
Regional predictors				
Socio-geographical index	40.5	19.3	4.1	17.8
Demographic index	1.7	0.5	0.4	0.1
Industrial index	1.5	0.1	0.2	0.4
Mixed index	7.7	11.5	10.9	12.3
Economic index	8.1	2.2	2.6	4.0

## Data Availability

The datasets presented in this article are not readily available because of the prohibition of data transfer to third parties. Requests to access the datasets should be directed to Svetlana A. Shalnova, svetlanashalnova@yandex.ru.

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
