# Peer review of "Mental Health of the Russian Federation Population versus Regional Living Conditions and Individual Income"

_ijerph, 2023, doi:10.3390/ijerph20115973_

Round 1

Reviewer 1 Report

Dear authors,

Thank you for the opportunity to read your extensive research.

The following are comments on each section of the manuscript:

Introduction. The detailed analysis of the studies is carried out, the aim of the study and a particular task are defined. The research hypotheses that the authors test in the work (which would help to understand the logic of the presentation of the results) are not indicated.

Materials and methods.

Please specify in connection with which the monitoring data of 2013-2014 are used? How up-to-date will this information be? In this connection, the authors have not conducted an additional, at least a small analysis recently (especially since these results could have changed significantly during the pandemic)? However, the authors do not indicate this in the study limitations.

Please specify in connection with what exactly these methods were chosen during statistical data processing, and other multivariate methods were not used?

Results. The results are very overloaded, the information is difficult to perceive, it is difficult to highlight the key results of the study. Please make the description more logical, highlight before discussing the results which key results according to your goal and hypotheses you are defining.

The discussion of the results. The authors present similar results of studies performed on other samples. Limitations of the study are noted. At the same time, in the discussion, the authors give insufficient detail on the discussion of the results obtained, as well as how they can be applied.

As a result, the article is currently in need of revision.

Best regards, the reviewer

Author Response

We thank the reviewer for commenting on our article.

Let us address your comments:

Introduction. The detailed analysis of the studies is carried out, the aim of the study and a particular task are defined. The research hypotheses that the authors test in the work (which would help to understand the logic of the presentation of the results) are not indicated.

Response: The research hypothesis has been added to the Introduction (Lines 91-94).

Please specify in connection with which the monitoring data of 2013-2014 are used? How up-to-date will this information be? In this connection, the authors have not conducted an additional, at least a small analysis recently (especially since these results could have changed significantly during the pandemic)? However, the authors do not indicate this in the study limitations.

Response: We conducted analysis of monitoring results for 2013-2014 because these are the most recent epidemiological data including individual information about the mental health of the Russian population in several regions of Russia at once (11 regions). A similar more recent epidemiological study in 2017 was performed in four regions that significantly reduces the possibility of regional characteristics assessment. In addition, in our opinion the analysis based on the data of 2013-2014 is of theoretical interest in environmental influence assessment on mental health. An analysis of such exposure during the pandemic period is undoubtedly interesting both as an independent study and in terms of comparison it with the pre-COVID-19 period. If we collect the necessary data, such analysis will undoubtedly be carried out. The comment has been added to the restrictions (Lines 385-390).

Please specify in connection with what exactly these methods were chosen during statistical data processing, and other multivariate methods were not used?

Response: The selection of statistical methods depends on research tasks and data type. When forming the mental health scale (MHS), the principal component method was used as one of the statistical methods for reducing the dimensionality of data. Generalized Estimating Equations (GEE) were chosen due to the analysis of a complex two-level sample (individuals in regions) and due to need to adjust the standard errors in the regression analysis. When choosing a specific regression analysis model (within the GEE), linear regression was chosen as the most suitable for a quantitative outcome.

The results are very overloaded, the information is difficult to perceive, it is difficult to highlight the key results of the study. Please make the description more logical, highlight before discussing the results which key results according to your goal and hypotheses you are defining.

Response: The Results describes the main associations of analysis. For a better understanding of the key results the first paragraph from the Discussion has been moved to the end of the Results (Lines 265-279). This paragraph just summarizes the main results of the study allowing to highlight the main ones.

The discussion of the results. The authors present similar results of studies performed on other samples. Limitations of the study are noted. At the same time, in the discussion, the authors give insufficient detail on the discussion of the results obtained, as well as how they can be applied.

Response: A paragraph with discussing the results obtained directly has been added as well as assumptions on how they can be applied (Lines 364-376).

In addition, we want to note that, in accordance with the editor's remark, the article text was reviewed for a similarity with other sources. We changed the wording in sections with the highest similarity with source #1: 2.1, 2.3, 2.4, 2.5. Some wording changes have been made in other article sections. Some similarities which are stable standard expressions (for example, health risk factors), index names, institution names, and technical phrases (for example, Informed Consent Statement) were left unchanged.

Reviewer 2 Report

This study was to assess the impact of regional living conditions on the 9 Russian population mental health. The results provided new fundamental 20 knowledge on the impact of the living environment on health. Here is my comment.

 1.     In line 189, is there any particular reason to use the Kruskal-Wallis test instead of the t test?

2.     In line 191, the authors mentioned “a generalized estimating equation with robust standard errors, taking into account the nested data structure (study subjects in regions, n=11), was used to measure associations.” However, I can't see any analysis results for GEE. In my opinion these models are derived from multiple regression and not so-called GEE.

3.     In line 127, the authors mentioned “to fulfill such assessment, we used the principal component method for three indicators to identify a latent variable with a high load on depression (0.81), stress (0.82), anxiety (0.86) and providing an explanation for 69.4% of the variance.” Therefore, this article will have 4 highly similar regression models analyzed. I don't think it is necessary to use PCA, to create a similar dependent variable (MHS). It can also be found from Table 2-5 that the results of these models are actually quite similar.

4.     Please write clearly, what statistical analysis does Table 6 come from? What is the purpose?

Author Response

We thank the reviewer for commenting on our article.

Let us address your comments:

  1. In line 189, is there any particular reason to use the Kruskal-Wallis test instead of the t test?

Response: The distribution of quantitative variables is different from normal. In this regard, it was decided to use not the Student's t-test which requires the normal distribution of the trait but its non-parametric analogue.

  1. In line 191, the authors mentioned “a generalized estimating equation with robust standard errors, taking into account the nested data structure (study subjects in regions, n=11), was used to measure associations.” However, I can't see any analysis results for GEE. In my opinion these models are derived from multiple regression and not so-called GEE.

Response: The GEE results look the same as the results of a conventional regression analysis that is the B-coefficient and level of its statistical significance. A GEE feature is the correction of the standard error in the regression which makes it possible to state with confidence that the grouping of individuals into regions did not affect the obtained associations.

  1. In line 127, the authors mentioned “to fulfill such assessment, we used the principal component method for three indicators to identify a latent variable with a high load on depression (0.81), stress (0.82), anxiety (0.86) and providing an explanation for 69.4% of the variance.” Therefore, this article will have 4 highly similar regression models analyzed. I don't think it is necessary to use PCA, to create a similar dependent variable (MHS). It can also be found from Table 2-5 that the results of these models are actually quite similar.

Response: Indeed, regression analyzes for depression, stress, anxiety, and mental health showed broadly similar results. However, we find it useful to show not only depression, stress, and anxiety separately but also the combination of these three indicators into a single summary indicator of mental health. The similarity of the results is explained by the single nature of depression, stress, and anxiety that is also reflected in the high factor load of these indicators with a latent variable (mental health). Many studies operate with only one of the indicators which raises the question of how comparable the patterns of environmental influences of depression, stress, and anxiety are. In our study among other things, it is shown that the comparability is a quite high.

  1. Please write clearly, what statistical analysis does Table 6 come from? What is the purpose?

Response: The contribution of regional and individual characteristics to mental health indicators is taken from regression models to better represent the importance of predictors. For a better understanding of this in the Methods (Lines 183-185) and in the title of Table 6 (Line 296) the phrase that the contribution is taken from regression models has been added. In addition, when describing the results of Table 6, text has been added to clarify the purpose (Lines 261-264).

In addition, we want to note that, in accordance with the editor's remark, the article text was reviewed for a similarity with other sources. We changed the wording in sections with the highest similarity with source #1: 2.1, 2.3, 2.4, 2.5. Some wording changes have been made in other article sections. Some similarities which are stable standard expressions (for example, health risk factors), index names, institution names, and technical phrases (for example, Informed Consent Statement) were left unchanged.

Round 2

Reviewer 1 Report

Dear authors,

thank you for taking into account the recommendations for the manuscript. Now the content of your research is more understandable and the conclusions are more justified.

The article can be recommended for publication.

Best regards, reviewer

Reviewer 2 Report

I have no further questions.